# Synthetic Data from Diffusion Models Improves ImageNet Classification

**Shekoofeh Azizi, Simon Kornblith**[*]**, Chitwan Saharia**[*]**, Mohammad Norouzi**[*]**, David J. Fleet**

*Google DeepMind*                                      *shekazizi@google.com, davidfleet@google.com*

**Reviewed on OpenReview:** *https://openreview.net/forum?id=DlRsoxjyPm*

## Abstract

Deep generative models are becoming increasingly powerful, now generating diverse, high fidelity, photo-realistic samples given text prompts. Nevertheless, samples from such models have not been shown to significantly improve model training for challenging and well-studied discriminative tasks like ImageNet classification. In this paper we show that augmenting the ImageNet training set with samples from a generative diffusion model can yield substantial improvements in ImageNet classification accuracy over strong ResNet and Vision Transformer baselines. To this end we explore the fine-tuning of large-scale text-to-image diffusion models, yielding class-conditional ImageNet models with state-of-the-art FID score (1.76 at $256 \times 256$ resolution) and Inception Score (239 at $256 \times 256$). The model also yields a new state-of-the-art in Classification Accuracy Scores, i.e., ImageNet test accuracy for a ResNet-50 architecture trained solely on synthetic data (64.96 top-1 accuracy for $256 \times 256$ samples, improving to 69.24 for $1024 \times 1024$ samples). Adding up to three times as many synthetic samples as real training samples consistently improves ImageNet classification accuracy across multiple architectures.

## 1 Introduction

Deep generative models are becoming increasingly mature to the point that they can generate high fidelity photo-realistic samples (Dhariwal & Nichol, 2021; Ho et al., 2020; Sohl-Dickstein et al., 2015). Most recently, denoising diffusion probabilistic models (DDPMs) (Ho et al., 2020; Sohl-Dickstein et al., 2015) have emerged as a new category of generative techniques that are capable of generating images comparable in quality to generative adversarial networks (GANs) while introducing greater stability during training (Dhariwal & Nichol, 2021; Ho et al., 2022b). This has been shown both for class-conditional generative models on classification datasets, and for open vocabulary text-to-image generation (Nichol et al., 2021; Ramesh et al., 2022; Rombach et al., 2022; Saharia et al., 2022b).

It is therefore natural to ask whether current models are powerful enough to generate natural image data that are effective for challenging discriminative tasks; i.e., generative data augmentation. Specifically, are diffusion models capable of producing image samples of sufficient quality and diversity to improve performance on well-studied benchmark tasks like ImageNet classification? ImageNet classification sets an exceptionally high standard due to the extensive tuning of existing architectures, augmentation strategies, and training recipes. A closely related question is, to what extent large-scale text-to-image models can serve as good representation learners or foundation models for downstream tasks? We explore this issue in the context of generative data augmentation, showing that these models can be fine-tuned to produce state-of-the-art class-conditional generative models on ImageNet.

To this end, we demonstrate three key findings. We show that the Imagen text-to-image model, when fine-tuned on the ImageNet training set produces state-of-the-art class-conditional ImageNet models at multiple resolutions, according to their Fréchet Inception Distance (FID) (Heusel et al., 2017) and Inception Score

---

[*]Work done at Google DeepMind.

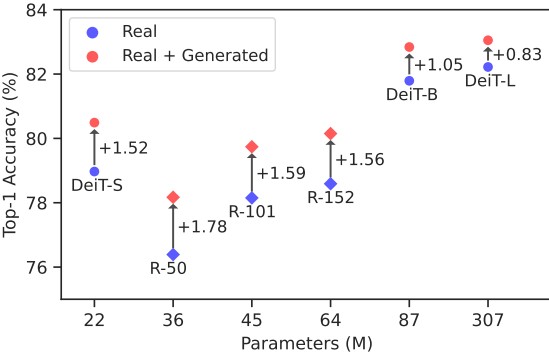 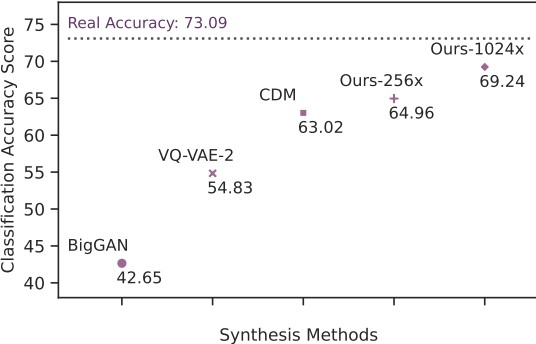

Figure 1: Left: Augmenting real training data with generated 1024×1024 images from our ImageNet model boosts classification accuracy for ResNet and Transformer models. Right: Classification Accuracy Scores (Ravuri & Vinyals, 2019) show models trained on generated data approaching those trained on real data.

(IS) (Salimans et al., 2016); e.g., we obtain an FID of 1.76 and IS of 239 on 256×256 image samples. These models outperform existing state-of-the-art models, with or without the use of guidance to improve model sampling. We further establish that data from such fine-tuned class-conditional models also provide new state-of-the-art Classification Accuracy Scores (CAS) (Ravuri & Vinyals, 2019), computed by training ResNet-50 models on synthetic images and then evaluating them on the real ImageNet validation set (Fig. 1 - right). Finally, we show that performance of models trained on generative data further improves by combining synthetic data with real data, with large amounts of synthetic data, and with longer training times. These results hold across a host of convolutional and Transformer-based architectures (Fig. 1 - left).

## 2 Related Work

**Synthetic Data.** The use of synthetic data has been widely explored for generating large amounts of labeled data for vision tasks that require extensive annotation. Examples include tasks like semantic image segmentation (Baranchuk et al., 2021; Chen et al., 2019; Li et al., 2022b; 2021; Sankaranarayanan et al., 2018; Tritrong et al., 2021), optical flow estimation (Dosovitskiy et al., 2015; Kim, 2022; Sun et al., 2021), human motion understanding (Guo et al., 2022; Izadi et al., 2011; Ma et al., 2022; Varol et al., 2017), and other dense prediction tasks (Baranchuk et al., 2021; Xu et al., 2021). Previous work has also explored 3D-rendered datasets (Greff et al., 2022; Zheng et al., 2020) and simulation environments with physically realistic dynamics engines (de Melo et al., 2021; Dosovitskiy et al., 2017; Gan et al., 2021).

Unlike methods that use model-based rendering, here we focus on the use of data-driven generative models of natural images, for which GANs have remained the predominant approach to date (Brock et al., 2019; Gowal et al., 2021; Li et al., 2022b). Nevertheless, models trained on data from BigGAN-deep (Brock et al., 2019) underperform cascaded diffusion models (Ho et al., 2022b) in classification accuracy score. In terms of image quality, BigDatasetGAN (Li et al., 2022a) FID scores underperform diffusion models, and neither DatasetGAN nor BigDatasetGAN report improved classifier performance with generated data on ImageNet over models trained solely on real data.

**Distillation and Transfer.** In our work, we use a diffusion model that has been pretrained on a large multimodal dataset and fine-tuned on ImageNet to provide synthetic data for a classification model. This setup has connections to previous work that has directly trained classification models on large-scale datasets and then fine-tuned them on ImageNet (Kolesnikov et al., 2020; Mahajan et al., 2018; Radford et al., 2021; Sun et al., 2017; Zhai et al., 2022). It is also related to knowledge distillation (Bucilă et al., 2006; Hinton et al., 2015) in that we transfer knowledge from the diffusion model to the classifier; although it differs from the traditional distillation setup in that we transfer this knowledge through generated data rather than labels. Our goal in this work is to show the viability of this kind of generative knowledge transfer with modern diffusion models.

**Diffusion Model Applications.** Diffusion models have been successfully applied to image generation (Ho et al., 2020; 2022b; Ho & Salimans, 2022), speech generation (Chen et al., 2020; Kong et al., 2020), and video generation (Ho et al., 2022a; Singer et al., 2022; Villegas et al., 2022), and have found applications in various image processing areas, including image colorization, super-resolution, inpainting, and semantic editing (Saharia et al., 2022a;c; Song et al., 2020; Wang et al., 2022). One notable application of diffusion models is large-scale text-to-image generation. Several text-to-image models including Stable Diffusion (Rombach et al., 2022), DALL-E 2 (Ramesh et al., 2022), Imagen (Saharia et al., 2022b), eDiff (Balaji et al., 2022), and GLIDE (Nichol et al., 2021) have produced evocative high-resolution images. However, the use of large-scale diffusion models to support downstream tasks is still in its infancy.

Very recently, large-scale text-to-image models have been used to augment training data. He et al. (2022) show that synthetic data generated with GLIDE (Nichol et al., 2021) improves zero-shot and few-shot image classification performance. They further pretrain a classifier on millions of examples generated using CIFAR-100 class names as prompts, and then fine-tune the classifier on the CIFAR-100 training set. The resulting classifier performs better than one trained from scratch on CIFAR-100. However, CIFAR-100 is simpler and much smaller than ImageNet, and many techniques that improve accuracy on CIFAR-100 fail to do so on ImageNet (e.g., Shake-Shake regularization (Gastaldi, 2017) and Cutout (DeVries & Taylor, 2017)). Trabucco et al. (2023) explore strategies to augment individual images using a pretrained diffusion model, demonstrating improvements in few-shot settings. StableRep (Tian et al., 2023) uses Stable Diffusion (Rombach et al., 2022) to demonstrate that if a generative model is configured with the correct classifier-free guidance scale, training self-supervised methods for contrastive representation learning on synthetic images can be as good or better than training them on real images.

Most closely related to our work, two recent papers train ImageNet classifiers on images generated by diffusion models, although they explore only the pretrained Stable Diffusion model and do not fine-tune it (Bansal & Grover, 2023; Sariyildiz et al., 2022). Bansal & Grover (2023) find that a model trained on a combination of the ImageNet training set and synthetic data generated by base or fine-tuned text-to-image diffusion models yields similar or worse accuracy on the ImageNet validation accuracy than a model trained solely on the ImageNet training set. We instead find that, with an appropriate choice of fine-tuning hyper-parameters, one can achieve significant performance gains by adding synthetic data from a fine-tuned model. Sariyildiz et al. (2022) train an ImageNet classifier with samples from a text-to-image diffusion model, reporting 42.9% Top-1 accuracy. In contrast, with samples from our fine-tuned model we obtain SOTA Top-1 test accuracy, between 69% to 75%, on standard benchmark architectures and training procedures.

## 3    Background

**Diffusion.** Diffusion models work by gradually destroying the data through the successive addition of Gaussian noise, and then learning to recover the data by reversing this noising process (Ho et al., 2020; Sohl-Dickstein et al., 2015). In broad terms, in a forward process random noise is slowly added to the data as time $t$ increases from $0$ to $T$. A learned reverse process inverts the forward process, gradually refining a sample of noise into an image. To this end, samples at the current time step, $x_{t-1}$, are drawn from a learned Gaussian distribution, $\mathcal{N}(x_{t-1}; \mu_\theta(x_t, t), \Sigma_\theta(x_t, t))$, where the mean of the distribution $\mu_\theta(x_t, t)$, is conditioned on the sample at the previous time step. The variance $\Sigma_\theta(x_t, t)$ follows a fixed schedule. In conditional diffusion models, the reverse process is associated with a conditioning signal, such as a class label in class-conditional models (Ho et al., 2022b).

Diffusion models have been the subject of many recent papers including important innovations in architectures and training (e.g., (Balaji et al., 2022; Ho et al., 2022b; Nichol & Dhariwal, 2021; Saharia et al., 2022b)). Important below, Ho et al. (2022b) propose cascades of diffusion models at increasing image resolutions for high resolution images. Other work has explored the importance of the generative sampling process, introducing new noise schedules, guidance mechanisms to trade-off diversity with image quality, distillation for

efficiency, and different parameterizations of the denoising objective (e.g., (Hoogeboom et al., 2023; Karras et al., 2022; Saharia et al., 2022b; Salimans & Ho, 2022)).

**Classification Accuracy Score.** It is a standard practice to use FID (Heusel et al., 2017) and Inception Score (Salimans et al., 2016) to evaluate the visual quality of generative models. Due to their relatively low computation cost, these metrics are widely used as proxies for generative model training and tuning. However, both methods tend to penalize non-GAN models harshly, and Inception Score produces overly optimistic scores in methods with sampling modifications (Ho & Salimans, 2022; Ravuri & Vinyals, 2019). Ravuri & Vinyals (2019) also argued that these metrics do not show a consistent correlation with metrics that assess performance on downstream tasks like classification accuracy.

An alternative way to evaluate the quality of samples from generative models is to examine the performance of a classifier that is trained on generated data and evaluated on real data (Santurkar et al., 2018; Yang et al., 2017). To this end, Ravuri & Vinyals (2019) propose classification accuracy score (CAS), which measures classification performance on the ImageNet validation set for ResNet-50 models (He et al., 2016) trained on generated data. It is an intriguing proxy, as it requires generative models to produce high fidelity images across a broad range of categories, competing directly with models trained on real data.

To date, CAS performance has not been particularly compelling. Models trained exclusively on generated samples underperform those trained on real data. Also, performance drops when even relatively small amounts of synthetic data are added to real data during training (Ravuri & Vinyals, 2019). This performance drop may arise from issues with the quality of generated samples, their diversity (e.g., due to mode collapse in GAN models), or both. Cascaded diffusion models (Ho et al., 2022b) were recently shown to outperform BigGAN-deep (Brock et al., 2019) and VQ-VAE-2 (Razavi et al., 2019) on CAS. That said, there remains a sizeable gap in ImageNet test performance between models trained on real data and those trained on synthetic data (Ho et al., 2022b). Here, we explore the use of diffusion models in greater depth, with much stronger results, demonstrating the advantage of large-scale models and fine-tuning.

## 4 Generative Model Training and Sampling

In what follows we address two main questions: whether large-scale text-to-image models can be fine-tuned as class-conditional ImageNet models, and to what extent these models are useful for generative data augmentation. For this purpose, we undertake a series of experiments to construct and evaluate such models, focused primarily on data sampling for use in training ImageNet classifiers. The ImageNet ILSVRC 2012 dataset (ImageNet-1K) comprises 1.28 million labeled training images and 50K validation images spanning 1000 categories (Russakovsky et al., 2015). We adopt ImageNet-1K as our benchmark to assess the efficacy of the generated data, as this is one of the most widely and thoroughly studied benchmarks for which there is an extensive literature on architectures and training procedures, making it challenging to improve performance. Since the images of ImageNet-1K dataset vary in dimensions and resolution with the average image resolution of 469×387 (Russakovsky et al., 2015), we examine synthetic data generation at different resolutions, including 64×64, 256×256, and 1024×1024.

In contrast to previous work that trains diffusion models directly on ImageNet data (e.g., (Dhariwal & Nichol, 2021; Ho et al., 2022b; Hoogeboom et al., 2023)), here we leverage a large-scale text-to-image diffusion model (Saharia et al., 2022b) as a foundation, in part to explore the potential benefits of pre-training on a larger, generic corpus. A key challenge in doing so concerns the alignment of the text-to-image model with ImageNet classes. If, for example, one naively uses short text descriptors like those produced for CLIP by Radford et al. (2021) as text prompts for each ImageNet class, the data generated by the Imagen model (Saharia et al., 2022b) is easily shown to produce a poor ImageNet classifier. One problem is that a text label may be associated with multiple visual concepts in the wild, or visual concepts that differ systematically from ImageNet (see Fig. 2). This poor performance may also be a consequence of the high guidance weights used by Imagen, thereby sacrificing generative diversity for sample quality. While there are several ways in which one might re-purpose a text-to-image model as a class-conditional model, e.g., optimizing prompts to minimize distribution shifts, here we fix the prompts to be the associated CLIP class names (Radford et al., 2021), and fine-tune the weights and sampling parameters of the diffusion-based generative model.

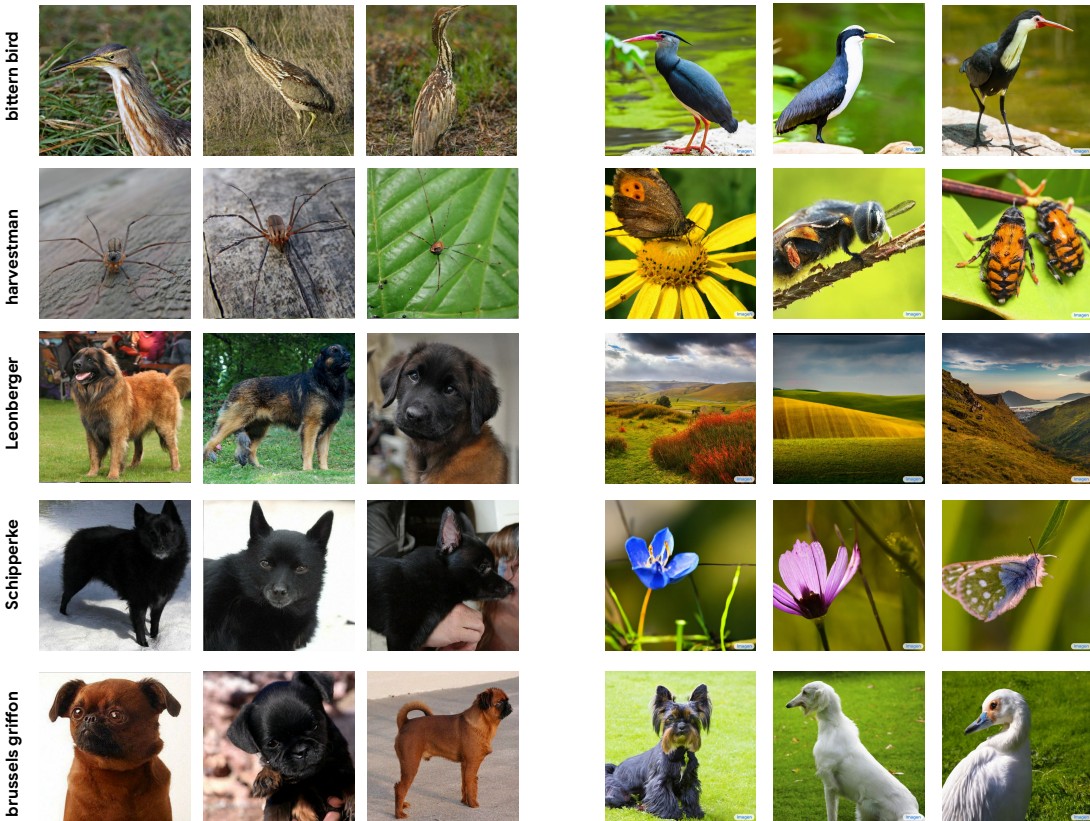

Figure 2: Example 1024×1024 images from the fine-tuned Imagen (left) model vs. vanilla Imagen (right). Fine-tuning and careful choice of guidance weights and other sampling parameters help to improve the alignment of images with class labels and sample diversity. More samples are provide in the Appendix.

## 4.1 Imagen Fine-tuning

We leverage the large-scale Imagen text-to-image model (Saharia et al., 2022b) as the backbone text-to-image generator that we fine-tune using the ImageNet training set. It includes a pretrained text encoder that maps text to contextualized embeddings, and a cascade of conditional diffusion models that map these embeddings to images of increasing resolution. Imagen uses a frozen T5-XXL encoder as a semantic text encoder to capture the complexity and compositionality of text inputs. The cascade begins with a 2B parameter 64×64 text-to-image base model. Its outputs are then fed to a 600M parameter super-resolution model to upsample from 64×64 to 256×256, followed by a 400M parameter model to upsample from 256×256 to 1024×1024. The base 64×64 model is conditioned on text embeddings via a pooled embedding vector added to the diffusion time-step embedding, like previous class-conditional diffusion models (Ho et al., 2022b). All three stages of the diffusion cascade include text cross-attention layers (Saharia et al., 2022b).

Given the relative paucity of high resolution images in ImageNet, we fine-tune only the 64×64 base model and 64×64→256×256 super-resolution model on the ImageNet-1K train split, keeping the final super-resolution module and text-encoder unchanged. The 64×64 base model is fine-tuned for 210K steps and the 64×64→256×256 super-resolution model is fine-tuned for 490K steps, on 256 TPU-v4 chips with a batch size of 2048. As suggested in the original Imagen training process, Adafactor (Shazeer & Stern, 2018) is used to fine-tune the base 64×64 model because it had a smaller memory footprint compared to Adam (Kingma & Ba, 2014). For the 256×256 super-resolution model, we used Adam for better sample quality. Throughout fine-tuning experiments, we select models based on FID score calculated over 10K samples from the default Imagen sampler and the ImageNet-1K validation set.

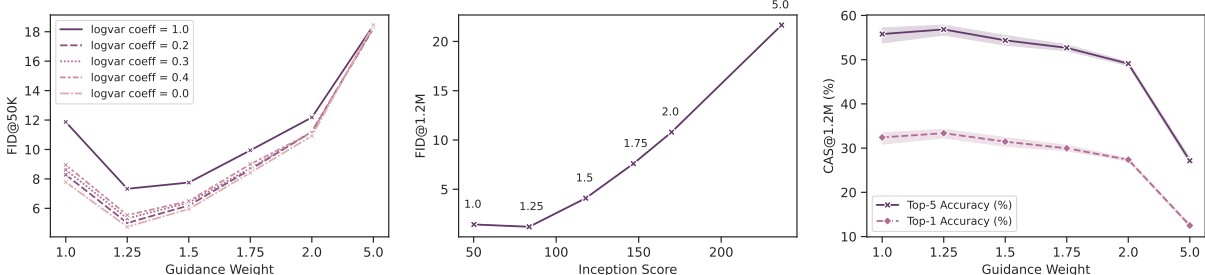

Figure 3: Sampling refinement for $64 \times 64$ base model. **Left**: Validation set FID vs. guidance weights for different values of log-variance. **Center**: Pareto frontiers for training set FID and IS at different values of the guidance weight. **Right**: Dependence of Classification Accuracy Score evaluated on the ImageNet training set (denoted CAS@1.2M) on guidance weight.

## 4.2 Sampling Parameters

The quality, diversity, and speed of text-conditioned diffusion model sampling are strongly affected by multiple factors including the number of diffusion steps, noise condition augmentation (Saharia et al., 2022b), guidance weights for classifier-free guidance (Ho & Salimans, 2022; Nichol et al., 2021), and the log-variance mixing coefficient used for prediction (Eq. 15 in (Nichol & Dhariwal, 2021)), described in further detail in Appendix A.1. We conduct a thorough analysis of the dependence of FID, IS and classification accuracy scores (CAS) in order to select good sampling parameters for the downstream classification task.

The sampling parameters for the $64 \times 64$ based model establish the overall quality and diversity of image samples. We first sweep over guidance weight, log-variance, and number of sampling steps, to identify good hyperparameters based on FID-50K (vs. the ImageNet validation set). Using the DDPM sampler (Ho et al., 2020) for the base model, we sweep over guidance values of $[1.0, 1.25, 1.5, 1.75, 2.0, 5.0]$ and log-variance of $[0.0, 0.2, 0.3, 0.4, 1.0]$, and denoise for 128, 500, or 1000 steps. The results of this sweep, summarized in Figure 3, suggest that optimal FID is obtained with a log-variance of 0 and 1000 denoising steps. Given these parameter choices we then complete a more compute intensive sweep, sampling 1.2M images from the fine-tuned base model for different values of the guidance weights. We measure FID on 50K samples from the validation set, and IS and CAS using the 1.2M ImageNet training set (denoted CAS@1.2M), to select the guidance weight for the base model sampler. Figure 3 shows the Pareto frontiers for FID vs. IS across different guidance weights, as well as the dependence of CAS on guidance weight, suggesting that optimal FID and CAS are obtained at a guidance weight of 1.25.

Given $64 \times 64$ samples obtained with the optimal hyperparameters, we then analyze the impact of guidance weight, noise augmentation, and log-variance to select sampling parameters for the super-resolution models. The noise augmentation value specifies the level of noise augmentation applied to the input to super-resolution stages in the Imagen cascade to regulate sample diversity (and improve robustness during model training). Here, we sweep over guidance values of $[1.0, 2.0, 5.0, 10.0, 30.0]$, noise conditioning augmentation values of $[0.0, 0.1, 0.2, 0.3, 0.4]$, and log-variance mixing coefficients of $[0.1, 0.3]$, and denoise for 128, 500, or 1000 steps. Figure 4 shows Pareto curves of FID Train vs. CAS@1.2M (i.e., for classification evaluated on the real training data) for the $64 \times 64 \rightarrow 256 \times 256$ super-resolution module across different noise conditioning augmentation values using a guidance weight of 1.0. These curves demonstrate the combined impact of the log-variance mixing coefficient and condition noise augmentation in achieving an optimal balance between FID and CAS.

Overall, the results suggest that FID and CAS are highly correlated, with smaller guidance weights leading to better CAS but negatively affecting Inception Score. We observe that using noise augmentation of 0 yields the lowest FID score for all values of guidance weights for super-resolution models. Nevertheless, it is worth noting that while larger amounts of noise augmentation tend to increase FID, they also produce more diverse samples, as also observed by Saharia et al. (2022b). Results of these studies are available in the Appendix.

Based on these sweeps, taking FID and CAS into account, we selected guidance of 1.25 when sampling from the base model, and 1.0 for other resolutions. We use DDPM sampler (Ho et al., 2020) log-variance mixing

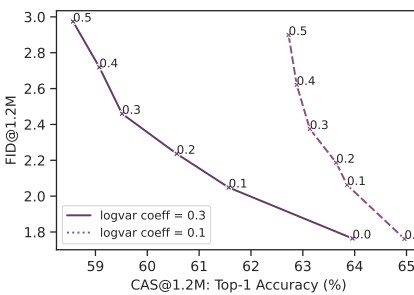 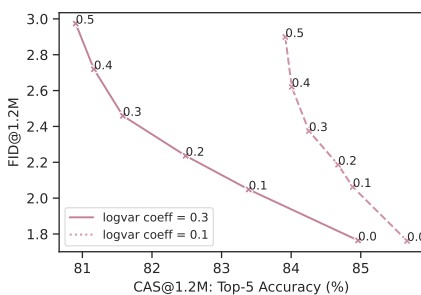

Figure 4: Training set FID vs. classification top-1 and top-5 accuracy Pareto curves under varying noise conditions when the guidance weight is set to 1.0 for resolution 256×256. These curves depict the joint influence of the log-variance mixing coefficient (Nichol & Dhariwal, 2021) and noise conditioning augmentation (Ho et al., 2022b) on FID Train and and CAS@1.2M.

coefficients of 0.0 for $64 \times 64$ samples, and 0.1 for $256 \times 256$ samples, with 1000 denoising steps. At resolution $1024 \times 1024$ we use a DDIM sampler (Song et al., 2021) with 32 steps, as in (Saharia et al., 2022b). We do not use noise conditioning augmentation for sampling.

### 4.3 Generation Protocol

We use the fine-tuned Imagen model with the optimized sampling hyper-parameters to generate synthetic data resembling the training split of ImageNet dataset. This means that we aim to produce the same quantity of images for each class as found in the real ImageNet dataset while keeping the same class balance as the original dataset. We then constructed large-scale training datasets with ranging from 1.2M to 12M images, i.e., between $1\times$ to $10\times$ the size of the original ImageNet training set.[1]

## 5 Results

### 5.1 Sample Quality: FID and IS

Despite the shortcomings described in Sec. 3, FID (Heusel et al., 2017) and Inception Score (Salimans et al., 2016) remain standard metrics for evaluating generative models. Table 1 reports FID and IS for our approach and existing class-conditional and guidance-based approaches.[2] Our fine-tuned model outperforms all the existing methods, including state-of-the-art methods that use U-Nets (Ho et al., 2022b) and larger U-ViT models trained solely on ImageNet data (Hoogeboom et al., 2023). This suggests that large-scale pretraining followed by fine-tuning on domain-specific target data is an effective strategy to achieve better visual quality with diffusion models, as measured by FID and IS. Figure 2 shows imaage samples from the fine-tuned model (see the Appendix for more samples). Note that our state-of-the-art FID and IS on ImageNet are obtained without any design changes, i.e., by simply adapting an off-the-shelf, diffusion-based text-to-image model to new data through fine-tuning. This is a promising result indicating that in a resource-limited setting, one can improve the performance of diffusion models by fine-tuning model weights and adjusting sampling parameters.

### 5.2 Classification Accuracy Score

As noted above, classification accuracy score (CAS) (Ravuri & Vinyals, 2019) is a better proxy than FID and IS for performance of downstream training on generated data. CAS measures ImageNet classification accuracy on the real test data for models trained solely on synthetic samples. In keeping with the CAS

---

[1]Imagen was trained on a mixture of datasets, 30% of which comprised Laion-400M (Schuhmann et al., 2021). We note that approximately 1.02% of ImageNet can be found in Laion-400M, which is relevant when interpreting classification accuracy (Cherti et al., 2023). In line with previous work, Cherti et al. (2023) also conclude that duplication in test sets does not significantly alter downstream results of ImageNet models.

[2]Because the ImageNet validation set was used for sampling hyper-parameter selection, our FID validation quantities may not be directly comparable to other models, but we report it here for completeness.

Table 1: Comparison of sample quality of synthetic ImageNet datasets measured by FID and Inception Score (IS) between our fine-tuned Imagen model and generative models in the literature. We achieve SOTA FID and IS on ImageNet generation among other existing models, including class-conditional and guidance-based sampling without any design changes.

| Model | FID train | FID validation | IS |
|---|---|---|---|
| 64x64 resolution | | | |
| BigGAN-deep (Dhariwal & Nichol, 2021) | 4.06 | - | - |
| Improved DDPM (Nichol & Dhariwal, 2021) | 2.92 | - | - |
| ADM (Dhariwal & Nichol, 2021) | 2.07 | - | - |
| CDM (Ho et al, 2022) | 1.48 | 2.48 | $67.95 \pm 1.97$ |
| RIN (Jabri et al., 2022) | 1.23 | - | 66.5 |
| RIN + noise schedule (Chen, 2023) | 2.04 | - | 55.8 |
| **Ours** (Fine-tuned Imagen) | 1.21 | 2.51 | $85.77 \pm 0.06$ |
| 256x256 resolution | | | |
| BigGAN-deep (Brock et al., 2019) | 6.9 | - | $171.4 \pm 2.00$ |
| VQ-VAE-2 (Razavi et al., 2019) | 31.11 | - | - |
| SR3 (Saharia et al., 2021) | 11.30 | - | - |
| LDM-4 (Rombach et al., 2022) | 10.56 | - | 103.49 |
| DiT-XL/2 (Peebles & Xie, 2022) | 9.62 | - | 121.5 |
| ADM (Dhariwal & Nichol, 2021) | 10.94 | - | 100.98 |
| ADM+upsampling (Dhariwal & Nichol, 2021) | 7.49 | - | 127.49 |
| CDM (Ho et al, 2022) | 4.88 | 3.76 | $158.71 \pm 2.26$ |
| RIN (Jabri et al., 2022) | 4.51 | - | 161.0 |
| RIN + noise schedule (Chen, 2023) | 3.52 | - | 186.2 |
| Simple Diffusion (U-Net) (Hoogeboom et al., 2023) | 3.76 | 2.88 | $171.6 \pm 3.07$ |
| Simple Diffusion (U-ViT L) (Hoogeboom et al., 2023) | 2.77 | 3.23 | $211.8 \pm 2.93$ |
| **Ours** (Fine-tuned Imagen) | 1.76 | 2.81 | $239.18 \pm 1.14$ |

Table 2: Classification Accuracy Scores (CAS) for 256×256 and 1024×1024 generated samples. CAS for real data and other models are obtained from (Ravuri & Vinyals, 2019) and (Ho et al., 2022b). Results indicate that the fine-tuned generative diffusion model outperforms previous methods by a substantial margin.

| Model | Top-1 Accuracy (%) | Top-5 Accuracy(%) |
|---|---|---|
| Real | 73.09 | 91.47 |
| BigGAN-deep (Brock et al., 2019) (Brock et al., 2019) | 42.65 | 65.92 |
| VQ-VAE-2 (Razavi et al, 2019) (Razavi et al., 2019) | 54.83 | 77.59 |
| CDM (Ho et al, 2022) (Ho et al., 2022b) | 63.02 | 84.06 |
| **Ours** (256×256 resolution) | 64.96 | 85.66 |
| **Ours** (1024×1024 resolution) | 69.24 | 88.10 |

protocol (Ravuri & Vinyals, 2019), we train a standard ResNet-50 architecture on a single crop from each training image. Models are trained for 90 epochs with a batch size of 1024 using SGD with momentum (see Appendix A.4 for details). Regardless of the resolution of the generated data, for CAS training and evaluation, we resize images to 256×256 (or, for real images, to 256 pixels on the shorter side) and then take a 224×224 pixel center crop.

Table 2 reports CAS for samples from our fine-tuned models at resolutions 256×256 and 1024×1024. CAS for real data and for other models are taken from (Ravuri & Vinyals, 2019) and (Ho et al., 2022b). The results

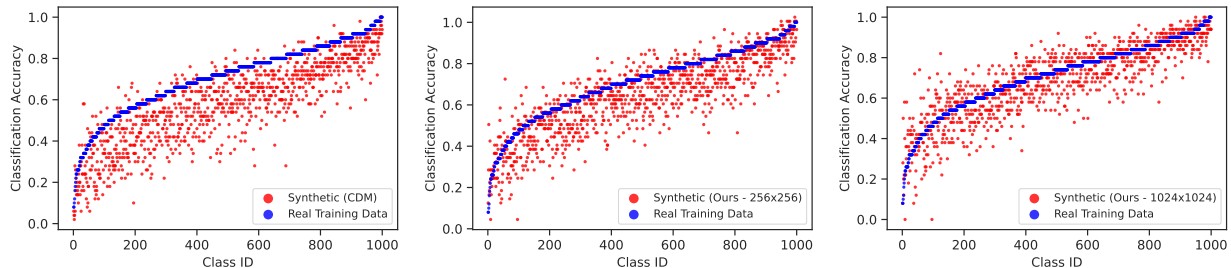

Figure 5: Class-wise classification accuracy comparison accuracy of models trained on real data (blue) and generated data (red). **Left**: The $256 \times 256$ CDM model (Ho et al., 2022b). **Middle and right:** Our fine-tuned class-conditional diffusion model at $256 \times 256$ and $1024 \times 1024$.

indicate that our fine-tuned class-conditional models outperform the previous methods in the literature at $256 \times 256$ resolution by a good margin, for both Top-1 and Top-5 accuracy. Interestingly, results are markedly better for $1024 \times 1024$ samples, even though these samples are down-sampled to $256 \times 256$ during classifier training. As reported in Table 2, we achieve the SOTA Top-1 classification accuracy score of 69.24% at resolution $1024 \times 1024$. This greatly narrows the gap with the ResNet-50 model trained on real data.

Figure 5 shows the accuracy of models trained on generative data (red) compared to a model trained on real data (blue) for each of the 1000 ImageNet classes (cf. Fig. 2 in (Ravuri & Vinyals, 2019)). From Figure 5 (left) one can see that the ResNet-50 trained on CDM samples is weaker than the model trained on real data, as most red points fall below the blue points. For our fine-tuned Imagen models (Figure 5 middle and right), however, there are more classes for which the models trained on generated data outperform the model trained on real data. This is particularly clear at $1024 \times 1024$.

## 5.3 Classification Accuracy with Different Models

To further evaluate the discriminative power of the synthetic data, compared to the real ImageNet-1K data, we analyze the classification accuracy of models with different architectures, input resolutions, and model capacities. We consider multiple ResNet-based and Vision Transformers (ViT)-based (Dosovitskiy et al., 2020) classifiers including ResNet-50 (He et al., 2016), ResNet-RS-50, ResNet-RS-152×2, ResNet-RS-350×2 (Bello et al., 2021), ViT-S/16 (Beyer et al., 2022), and DeiT-B (Touvron et al., 2021). The models trained on real, synthetic, and the combination of real and synthetic data, are all trained in the same way, consistent with the training recipes specified by authors of these models on ImageNet-1K. We verified that these methods trained on real data agree with the published results. The Appendix has more details on model training.

Table 3 reports the Top-1 validation accuracy of multiple ConvNet and Transformer models when trained with the 1.2M real ImageNet training images, with 1.2M generated $1024 \times 1024$ images, and when the generative samples are used to augment the real data. As one might expect, models trained solely on generated samples perform worse than models trained on real data. Nevertheless, augmenting real data with synthetic images from the diffusion model yields a substantial boost in performance across all classifiers tested; using a paired t-test, the results are statistically significant with $p = 10^{-5}$.

In addition, there is no obvious trend in the performance improvement of ConvNets versus Transformers as performance gains vary with several factors, including the number of model parameters and the resolution of the input image.

## 5.4 Merging Real and Synthetic Data at Scale

We next consider how performance of a ResNet-50 classifier depends on the amount of generated data that is used to augment the real data. Here we follow the conventional training recipe, training with random crops for 130 epochs, which yields higher ResNet-50 accuracy here than in the CAS results in Table 2 that used center crop and only 90 epochs. The Appendix provides training details.

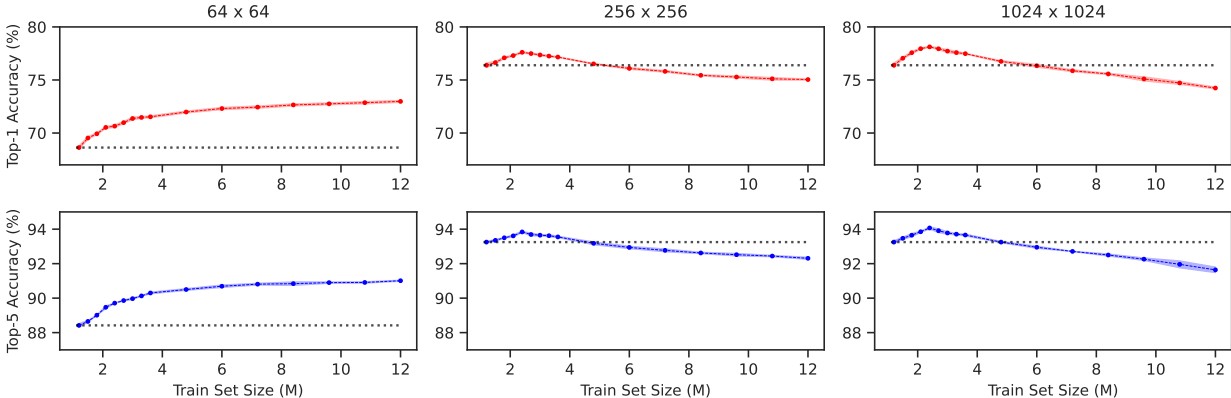

Figure 6: Top-1 and Top-5 classification accuracy of a ResNet-50 architecture, trained with the union of the real ImageNet Train data and increasing numbers of synthetic images at 64×64 (left), 256×256 (middle), and 1024×1024 (right) resolutions. We plot the mean accuracy and the standard deviation (depicted by shaded area) over 10 independently trained models. The leftmost point on all curves shows the accuracy of a model trained solely on the real ImageNet Train set, with other points showing increasing numbers of synthetic data, from 1× to 10× the size of the ImageNet Train set. (See Table A.3 for accuracy numbers.)

Ravuri & Vinyals (2019) (their Fig. 5) found that for almost all models tested, mixing generated samples with real data degrades Top-5 classifier accuracy. For Big-GAN-deep (Brock et al., 2019) with low truncation values (sacrificing diversity for sample quality), accuracy increases marginally with small amounts of generated data, but then quickly drops below models trained solely on real data when the amount of generated data approaches the size of the real train set. By comparison, Figure 6 shows that, for 64×64 images, performance continues to improve as the amount of generated data increases up to nine times the amount of real data, to a total dataset size of 12M images. Performance with samples from the two higher resolution models increases with up to 1.2M synthetic samples and then slowly decreases. Nevertheless, performance remains above the baseline model trained solely on real data for up to approximately 4M synthetic samples, i.e., with three times more synthetic data than real data. (Table A.3 in the Appendix provides these results in tabular form.)

## 6 Discussion and Future Work

This paper asks to what extent generative data augmentation is effective with current diffusion models. We do so in the context of ImageNet classification, a challenging task that has been studied extensively with well honed architectures and training recipes. Here we show that large-scale text-to-image diffusion models can be fine-tuned to produce class-conditional models with SOTA FID (1.76 at 256×256 resolution) and Inception Score (239 at 256×256). The resulting generative model also yields a new SOTA in Classification Accuracy Scores (64.96 at 256×256, improving to 69.24 for 1024×1024 samples). We also showed that improvements to ImageNet classification accuracy extend to large amounts of generated data, across a range of ResNet and Transformer-based models. Our work demonstrates the power of modern diffusion models for data generation and augmentation. However, several questions remain unanswered.

Although ImageNet classification is one of the most extensively studied benchmarks in vision and representation learning, many important tasks differ substantially from ImageNet, and the efficacy of generative data augmentation for such tasks remains unclear. Previous work has shown that accuracy on ImageNet is a good proxy for out-of-distribution accuracy (Taori et al., 2020) and transfer to other natural image tasks (Kornblith et al., 2019), but not for performance of architectures trained from scratch on small datasets (Tuggener et al., 2022) or for transfer to non-natural image tasks (Abnar et al., 2021; Fang et al., 2023). It thus remains crucial to validate our approach across various datasets and dataset sizes to establish the limits of its effectiveness.

Table 3: Comparison of classifier Top-1 Accuracy (%) performance when 1.2M generated images are used for generative data augmentation. Models trained solely on generated samples perform worse than models trained on real data. Nevertheless, augmenting the real data with data generated from the fine-tuned diffusion model provides a substantial boost in performance across many different classifiers.

| Model | Input Size | Params (M) | Real Only | Generated Only | Real + Generated | Performance Δ |
|---|---|---|---|---|---|---|
| | | | | ConvNets | | |
| ResNet-50 | 224×224 | 36 | 76.39 | 69.24 | 78.17 | +1.78 |
| ResNet-101 | 224×224 | 45 | 78.15 | 71.31 | 79.74 | +1.59 |
| ResNet-152 | 224×224 | 64 | 78.59 | 72.38 | 80.15 | +1.56 |
| ResNet-RS-50 | 160×160 | 36 | 79.10 | 70.72 | 79.97 | +0.87 |
| ResNet-RS-101 | 160×160 | 64 | 80.11 | 72.73 | 80.89 | +0.78 |
| ResNet-RS-101 | 190×190 | 64 | 81.29 | 73.63 | 81.80 | +0.51 |
| ResNet-RS-152 | 224×224 | 87 | 82.81 | 74.46 | 83.10 | +0.29 |
| | | | | Transformers | | |
| ViT-S/16 | 224×224 | 22 | 79.89 | 71.88 | 81.00 | +1.11 |
| DeiT-S | 224×224 | 22 | 78.97 | 72.26 | 80.49 | +1.52 |
| DeiT-B | 224×224 | 87 | 81.79 | 74.55 | 82.84 | +1.04 |
| DeiT-B | 384×384 | 87 | 83.16 | 75.45 | 83.75 | +0.59 |
| DeiT-L | 224×224 | 307 | 82.22 | 74.60 | 83.05 | +0.83 |

The diffusion model we use was pretrained on a dataset much larger than ImageNet. It is natural to ask whether a sufficiently powerful diffusion model trained solely on ImageNet will still provide improvements when used for generative data augmentation. There are also other ways to leverage large datasets to improve performance on downstream tasks, such as transfer learning or non-generative data augmentation approaches. Further research is required to gain a comprehensive understanding of how different methods can complement each other and work in synergy to improve overall training performance.

Finally, our experiments uncover a couple of unexpected phenomena worthy of further study. One concerns the boost in CAS at resolution $1024 \times 1024$, suggesting that the larger images capture more useful image structure than those at 256×256, even though the 1024×1024 images are downsampled to 256×256 before being center-cropped to $224 \times 224$ for input to ResNet-50. Another concerns the sustained gains in classification accuracy with large amounts of synthetic data at 64×64, while at higher resolutions gains are not monotonic (Figure 6 and Table A.3). It may be that there is less information at low resolutions for training, and hence a greater opportunity at low resolutions for augmentation with synthetic images. Performance at high resolutions increases up to 1M synthetic images and then slowly declines; this may indicate greater bias in the model at high resolutions, or the need for more sophisticated training methods with synthetic data. These issues remain topics of on-going research.

**Broader Impact Statement**

The research on generative data augmentation have the potential to positively impact variety of fields by fostering the development of more robust and privacy-preserving models. However, it is important to be aware of the potential challenges and drawbacks associated with synthetic data generation and generative models. Transparency and clear guidelines regarding the use of synthetic data in safety critical applications are essential to ensure ethical and responsible adoption. Like the original Imagen model, our fine-tuned variant is not publicly available, in part to protect against the generation of harmful content.

**Acknowledgments**

We thank Jason Baldridge and Ting Chen for their valuable feedback. We also extend thanks to William Chan, Saurabh Saxena, and Lala Li for helpful discussions, feedback, and support with the Imagen code.

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
