# OpenReview forum: "Synthetic Data from Diffusion Models Improves ImageNet Classification"
_TMLR — Accepted by TMLR_

### Review · Reviewer_XfEJ · 2023-08-12

**Summary Of Contributions:**

The authors dive into the question of whether diffusion-based synthetic images can augment classifier training and obtain better accuracy scores. They rely on a pre-trained Imagen, fine-tune it for ImageNet, optimize hyperparameters, and then use it to generate synthetic images and train a classifier based on them. The authors perform extensive experiments and ablation studies to assess the effect of synthetic data on classifier training, leading to improvements in classification accuracy for models trained on both real and synthetic data. Interestingly, models trained solely on synthetic data are not competitive with models trained solely on real data.

**Audience:**

Yes

**Claims And Evidence:**

Yes

**Requested Changes:**

Addressing the weaknesses listed above would be appreciated, especially for hyperparameter tuning based on training set instead of test set.

**Strengths And Weaknesses:**

Strengths:
- Very extensive experiments and ablation studies.
- Valid points, ideas, and methods for researching the question at hand.

Weaknesses:
- Hyperparameter tuning was done based on FID vs validation set. This fact greatly weakens the impact of the results (especially the SOTA FID claim), as the choices made for building the classifier include information leaked from the validation (test) set. I think that for hyperparameter tuning, it should have been done on a portion of the training set rather than the validation set (which is effectively the test set).
- How do we know that Imagen's large scale pre-training does not include images from the ImageNet validation set? It is important to protect against train-test contamination in papers like this one.
- Training on synthetic data still falls behind training on real data. Given that obtaining synthetic data is much more expensive, this does not seem like a great result (Table 2). Similarly, I am not sure that the performance deltas in Tables 3 and 4 are statistically significant.

---

> ### Author Response · Authors · 2023-09-09
>
> Thank you very much for your time and your feedback on the submission. The feedback has been taken into account in our revised paper. In what follows we respond to the comments and questions in the review.
>
> 1- **Use of ImageNet validation set for hyperparameter tuning:** We agree that,  because the ImageNet Val set was used for hyper-parameter search during fine-tuning of the generative model, the FID Val numbers need to be qualified.  We have added that qualification to the paper, but if it is deemed necessary, we could also remove the ‘FID Val’ column from Table 1, since FID Train (FID@1.2M) is the primary metric used in the literature. For clarity, it is also worth noting that all the CAS quantities used for hyper-parameter selection (i.e., as shown in Figures 3, 4, A.1, A.2, A.3, A.4, A.5, and A.6) measured classification accuracy on the (real) ImageNet Train set rather than the Val set.  We have revised the paper to clarify these points, and to avoid possible misunderstandings.
>
> 2- **Presence of ImageNet images in Imagen training set:** Imagen was trained on the mixture of two paired, image-text datasets, one of which is internal (proprietary) comprising 460M image-text pairs, and disjoint from ImageNet [1].  The other, LAION-400M, comprised 30% of the training data.  It has been reported that only 1.02% of ImageNet is found in LAION-400M [2], based on the use of pHash [3] to identify the overlap between LAION-400M and ImageNet.  Further, Cherti et al [2],  in line with multiple previous works, concludes that duplication in test sets does not significantly alter downstream results of ImageNet. We revised the paper to include this qualification, addressing train-test contamination.
>
> 3- **Lower accuracy for models trained solely on synthetic data than models trained solely on real data:** It is true that models trained solely on synthetic data underperform models trained solely on real data, as evidenced in prior research endeavors aimed at synthesizing datasets through diverse generative methods. Our CAS analysis shows this clearly in Table 2. Table 2 also demonstrates a significant improvement over the previous state-of-the-art generative models, showcasing a much closer alignment with real data (also see Figure 5).  That said, the goal of this paper is to demonstrate that by fine-tuning a large text-to-image diffusion model, one can obtain a generative model sufficiently strong that the synthetic data have a significant impact on discriminative training, even on very well-studied datasets like ImageNet. In our case, adding synthetic data to real data has a major impact on performance across a wide range of modern models used to benchmark ImageNet.  We feel this is a milestone for generative models that will certainly be of interest to the TMLR audience.
>
> 4- **Statistical significance of results:** Using a paired t-test, the results in Table 3 are significant with p = 10^{-5}. In Figure 6 and Table A.3 we show/report standard deviation over 10 independent training runs.  As such, dividing these quantities by sqrt(10) yields the standard error, again showing clear statistical significance. The revised paper clarifies these issues.
>
> [1] Saharia, Chitwan, et al. "Photorealistic text-to-image diffusion models with deep language understanding." NeurIPS 35 (2022): 36479-36494.
> [2] Cherti, Mehdi, et al. "Reproducible scaling laws for contrastive language-image learning." CVPR, 2023.
> [3] Zauner, Christoph. "Implementation and benchmarking of perceptual image hash functions." (2010).
>  [https://www.phash.org/docs/pubs/thesis_zauner.pdf]

---

### Review · Reviewer_iW4H · 2023-08-18

**Summary Of Contributions:**

Summary: The authors investigate the role of using synthetic data to improve classification performance on the ImageNet benchmark. In this context, the paper has two main contributions:

1) Large-scale text-to-image diffusion models like Imagen can be fine-tuned and adapted for class-conditional synthesis for ImageNet.

2) The resulting fine-tuned model can generate synthetic data that can be augmented with the original training dataset to improve ImageNet classification over neural net architectures like ResNet and Vision Transformer.

**Audience:**

Yes

**Broader Impact Concerns:**

Braoder Impact Statement has been specified by the authors.

**Claims And Evidence:**

Yes

**Requested Changes:**

Despite strong empirical results, I have a few questions:
1) Fig.1 looks great. However, it's unclear to me at which resolution was the synthetic data generated? It would be helpful to provide more clarity on this. Consequently, also for the results in Sec. 5.3

2) It is unclear how the final accuracy of the classifier (trained with real data augmented with generated data) varies with changes in the sampling parameters for the diffusion process. In the paper, the authors explore the optimal set of hyperparameters for achieving the best quality of generated samples and, therefore, (possibly) the best classification gains. However, an interesting question is if these gains in downstream classification performance are largely sensitive to small changes in the sampling hyperparameters. It would be great if the authors could provide some intuition or additional experimental results to verify this.

3) The results in Table 4 are interesting. The gain in classification accuracy decreases for higher resolutions as the proportion of generated data increases. However, this is not observed in Fig. 6 for 64x64 generated images. Does this mean the diffusion upsampler is not fine-tuned enough during training, or if using the upsampler leads to changes in the data distribution which makes the distribution of the updated train set (the original train set + large amounts of generated samples) different from the ImageNet test set, leading to poor accuracy scores? It would be great if the authors could provide some intuition on this. It might also be worth adding similar results as Table 4 for a transformer baseline (maybe only for 256x256) to check for architecture-related issues.

**Strengths And Weaknesses:**

Strengths:

1) Given the advances in the quality of synthetic data generation, this is a timely contribution.

2) The presentation of the experimental setup is clear. Experiments are extensive and justify using synthetic data for data augmentation to improve classification pipelines potentially.

---

> ### Author Response · Authors · 2023-09-09
>
> Thank you very much for your time and feedback on the submission.  The feedback has been taken into account in our revised paper.  In what follows we respond to the comments and questions in the review.
>
> 1- **Image resolution in Fig 1:**  The results shown in Figure 1 (left) and in Table 3 used generated image samples at resolution 1024 x 1024.  We have revised the paper to make this explicit. Thank you for catching the omission.
>
> 2- **Sensitivity to hyper-parameters:** The sensitivity of our ImageNet classifiers to the sampling hyper-parameters is explored in the context of the Classification Accuracy Score (eg, see Appendix Figures A.1, A.3, A.4, and A.6), showing smooth dependence of CAS of the hyper-parameters.  We did not do similar experiments with all models (eg, in Tables 3 and 4) because these experiments are expensive.  Instead, we used the CAS experiments with ResNet-50 and relatively small training times as a proxy for how other models might react. We can revise the paper to make this more explicit, but due to computation costs, we prefer not to repeat all these experiments for the other models.
>
> 3-**Differences in observations depending on image resolution:** The discrepancy between the behavior at low resolution and high resolution shown in Fig 6 is indeed interesting. At resolution 64x64, performance monotonically  improves with the amount of synthetic data we mix with real data.  At 1024x1024, performance improves over baseline for up to ~4M synthetic images, peaking close to 1.2M synthetic samples. As discussed in Conclusions, this remains a topic for further research. It may be that the divergence between the empirical data distribution and the model is greater at higher resolutions (e.g., due to diffusion-based super-resolution as the review suggests), or that downsampled training data is somewhat impoverished, yielding more room for improvement with synthetic data. These are interesting questions, but they will require more thought and experiments to explore thoroughly.  We have extended the discussion of this issue in the revised paper.

---

### Review · Reviewer_rkof · 2023-08-22

**Summary Of Contributions:**

My apologies to the authors for the slightly tardy review.

This paper evaluates the utility of using a large pretrained image generation model (ImageGen) with additional fine-tuning (on ImageNet-1k).

It evaluates the utility of this on two objectives, on sample quality (measured with Inception Score [IS] and the Frechet Inception Distance [FID]), and on the performance on downstream classification tasks when trained either on solely generated images, or on a mix of generated images and real data.

The authors demonstrate state of the are results on FID and IS on ImageNet-1k.

They also demonstrate state of the art results on Classification Accuracy Score (CAS) over a range of other generative models.

Finally they show that while training solely on generated data performs worse than training on real data, a balanced mix of real and generated data from the generative model can improve performance in classification accuracy over using real data only.

**Audience:**

Yes

**Broader Impact Concerns:**

No major concerns.

**Claims And Evidence:**

Yes

**Requested Changes:**

The changes I would request are addressed in the weaknesses.

Critical comments I would need to see discussed or made are 2,3,4.

Non-critical comments that would improve the paper in my estimation are 1,5,6. 6 in particular I think could make the paper significantly more interesting, particularly if one is interested in low-data regime applications, such as user specific personalisation.

**Strengths And Weaknesses:**

### Strengths
1. The paper is well written and easy to follow.
2. The detail of the experimental setup is exceptionally well done, as is the hyper parameter tuning.
3. The demonstrated results are clearly state of the art and interesting to the community.

### Weaknesses
1. I have a slight but not major nit-pick with the title. To me it seems the main result is that using _pretrained_ models on much larger datasets allows for artificial dataset augmentation, rather than using a diffusion model trained solely on the training data. While the distinction is narrow, I think it is important especially when first looking at the paper. I would prefer to see a qualifier such as "Synthetic Data from *Pretrained* Diffusion Models Improves ImageNet Classification" or similar.
2. In the literature review the authors quote "Also, performance drops when even relatively small amounts of synthetic data are added to real data during training (Ravuri & Vinyals, 2019)". In fact in this paper in limited circumstances the authors do see very moderate gains in performance in some circumstances (see fig. 5 e.g.). This would also seem like a relevant baseline to report in the experiments in section 5.4.
3. Table 4 and figure 6 in section 5.4 seem to be reporting results for the same experiment - why are they separated? I.e. figure 6 should be a column in table 4, or vice versa.
4. In table 4 the authors do not use less that 1x the original dataset size of synthetic data augmentation, although it looks like it would be optimal to use less that this from analysing the pattern in the results. Lower amounts of data should be investigated, and indeed this looks like it might be in line with the results of Ravuri & Vinyals, 2019 Section 4.4 Figure 5.
5. There are no baseline methods reported in section 5.4, although these must exists, or it would be sensible to try other non-diffusion generative models in the setting of using generated examples as augmentation data.
6. Did the authors consider ablating over the _size_ of the real training dataset used to fine tune the diffusion / train the classifiers / train the raw generative models? One might expect that this fine tuning method would be even more impactful if there was significantly less training data available to train on as the transfer learning would be more useful. As the size of the "real" dataset grows I would expect the transfer learning to be less impactful. For example one could repeat the experiments of section section 5.4 where only 600k, 300k, 100k, 1k, examples are used to finetune / be the "real" data in the classification training.
7. The code as far as I can tell is not public available for scrutiny, severely hampering reproducibility.

---

> ### Author Response · Authors · 2023-09-09
>
> Thank you very much for your time and feedback on the submission.  We have revised the paper to take the feedback into account. In what follows we respond to the comments and questions in the review.
>
> 1- **Title:**  Thank you for suggesting a change in title.  We had thought of several titles, including ‘pre-trained’, or ‘fine-tuned’  or ‘text-to-image’ etc.  They all seemed to provide more specificity, but at the cost of requiring some qualification. Eg while we use pre-trained models, we do not use them zero-shot.  We’ll continue to think of something more specific, but we are not sold on the inclusion of the word ‘pretrained’.  Though we would be receptive to other title suggestions.
>
> 2- **Baseline comparison with Fig 5 of Ravuri and Vinyals (2019):**  Our submission reports Top-1 accuracy, but we have added Top-5 accuracy for comparison to Fig 5 of Ravuri & Vinyals (2019) in Figure 6 and Table A.3. One can see that our Top-5 accuracy is significantly higher than all other models in their Fig 5.  Their best Top-5 accuracy across all models at 1.2M synthetic samples is roughly 93.0% while we obtain 94.07% at 100% synthetic data augmenting training dataset. They did not consider synthetic datasets any larger than this. We have also corresponded with Ravuri and their results and code are no longer available, so we cannot properly plot them in our paper.
>
> 3- **Merging results in Figure 6 and Table 4:**  This is a good idea. We decided to incorporate all resolutions into Figure 6, and we put Top-1 and Top-5 results into a table for completeness. This table was placed in the Appendix (ie Table A.3).
>
> 4- **Experiments to resolve peak performance in Table 4:**  Yes, it would be useful to add more points along the x-axis to help locate peak performance.  So we added more points Figure 6 and Table A.3.  The new results show performance with samples from the two higher resolution models increases with up to ~ 1.2M synthetic samples and then slowly decreases. Nevertheless, performance remains above the baseline model (ie trained solely on real data) for up to approximately 4M synthetic samples.
>
> 5- **Baselines for Section 5.4:**  We are not aware of other baselines for large-scale generative data augmentation for ImageNet classification.  The only studies we are aware of are those in Ravuri and Vinyals (2019) but those show substantial degradation in performance for most models even for relatively small generative datasets. We are not aware of any other paper reporting results like this on the full ImageNet-2012 datasets.
>
> 6-**Titrating the amount of real training data:**  This would be interesting to do, ie. training models with generated data augmenting different fractions of the ImageNet train set. Unfortunately, the suggestions about more points for Fig 6 and Table 4 is a higher priority from our perspective so we have not yet had the resources to train models with less real data.  We leave this to future work.
>
> 7- We are not planning to release our model weights, for fine-tuning the Imagen model, or our code for training ImageNet classifiers. Code for training ImageNet classifiers is readily available, and our code for fine-tuning the diffusion model is generic. Similar to the original Imagen, we do not plan to release the model weights out of concerns with safety and privacy, for the same reasons explained in depth in the Imagen paper (Saharia et al, NeurIPS 2022).

---

### Review · Reviewer_VjD7 · 2023-08-24

**Summary Of Contributions:**

This paper looks into the possibility of augmenting existing annotated datasets (Imagenet specifically) by T2I models to improve performance. They make two key contributions. First, they make an empirical study into the optimum way of fine-tuning large-scale T2I models like Imagen on specific datasets like ImageNet. Then, using the fine-tuned model, they generate more data to augment the original training set and show that augmentation using synthetic data improves performance.

**Audience:**

Yes

**Broader Impact Concerns:**

Adequately addressed

**Claims And Evidence:**

Yes

**Requested Changes:**

- I request the authors to add results from at least some of the aforementioned scenarios so we get a more in-depth insight into the working of the proposed method.

**Strengths And Weaknesses:**

Strengths
-----------------

- This paper adds to the increasing efforts aimed at leveraging the power of T2I models towards improving performance on a downstream classification task.

- The extensive study to find the optimum set of sampling parameters and fine-tuning strategies might be useful for future works.

- The observation that adding more and more generated data might start giving negative returns is a potentially valuable one for the community.

Weaknesses
---------------

The core contributions of the paper are unclear to me. If I understand correctly, this paper first conducts an extensive ablation on way to fine-tune Imagen on ImageNet dataset, and then shows that adding synthetic data to ImageNet improves ImageNet performance. While potentially valuable, I am not sure these results in themselves are enough to gain sufficient understanding of the method. Specifically, the following can still be potentially explored.

- Does the synthetic data which improve performance on ImageNet validation task also improve other properties like robustness and adaptation? The authors can conduct similar studies as in citation [Sariyildiz 2022] on other variants of ImageNet like ImageNet-A, ImageNet-V2 etc. to study these. More interestingly, is there an optimum way to generate synthetic data to improve the robustness and adaptation performance on these datasets?

- Do the benefits hold only for ImageNet? Say I want to improve performance on another dataset like Places-365, will I have to redo the whole sampling study experiments or will same observations/hyperparameters also help in generating data from Places which improve performance?

- Currently, the authors only explore end-to-end fine-tuning, which might be quite costly and intensive for most applications. Can the similar benefits be observed even when we use alternative low-cost fine-tuning strategies like adapter-tuning or prompt-tuning [1]?

- The observation that synthetic data yields negative returns is surprising, non-trivial and useful. Do the authors have some suggestion to overcome this bottleneck in future works?

- Also, the study only considers diffusion based generative models, it would also be useful (though unreasonable for a rebuttal timeline) to consider other types of generation methods like non-autoregressive models [2].

[1] Sohn, Kihyuk, et al. "Visual prompt tuning for generative transfer learning." Proceedings of the IEEE/CVF Conference on Computer Vision and Pattern Recognition. 2023.

[2] Chang, Huiwen, et al. "Maskgit: Masked generative image transformer." Proceedings of the IEEE/CVF Conference on Computer Vision and Pattern Recognition. 2022.

---

> ### Author Response · Authors · 2023-09-09
>
> Thank you very much for your time and your feedback on the submission.  We have revised the paper taking the feedback from the reviews into account.  In what follows we provide responses to your comments and questions in turn.
>
> First, to clarify our goals, this work aims to determine whether current diffusion models are capable of generating image samples of sufficient quality and diversity that, when used to augment real data, enable significant improvements in ImageNet classifiers beyond the performance of common benchmark architectures and training procedures. Despite the success of generative models in many contexts, attempts to do this with generative image models at large have not been successful to the best of our knowledge (e.g., Ravuri & Vinyals, 2019; Bansal & Grover 2023).  Improving ImageNet classification is a key milestone because ImageNet is a challenging task for which many model architectures and training procedures have been studied, at great length over many years, and it serves as a canonical baseline for new architectures and training methods. Although there are certainly many opportunities for further understanding of the effectiveness of synthetic data, we nonetheless believe that it will be of interest to TMLR’s audience that modern generative image models have finally achieved this milestone.
>
> Responses to specific comments and questions in the review:
>
>
> 1- **Robustness:**  Yes -- the extent to which OOD robustness is improved with synthetic data is interesting.  And as the reviewer points out, Sariyildiz (2022) already showed that synthetic data from diffusion models can enhance robustness. Whether generative data augmentation with our fine-tuned diffusion model improves OOD robustness is of interest, but it is not the primary question explored in our paper. Our primary concern is whether synthetic data improves ImageNet classification.  Sariyildiz et al (2022) reported 42.9% Top-1 accuracy on this task, well below performance with Big-GAN-deep and VQ-VAE-2 (as shown in Ravuri and Vinyals, 2019) and CDM (Ho et al, 2022); this is discussed in Section 2. While robustness is interesting, we prefer to maintain the focus of this paper on ImageNet classification performance per se.
>
> 2- **Do results extend to Places-365?**  ImageNet is a canonical baseline since many papers have shown that ImageNet performance is a good predictor of performance on a wide variety of other datasets. It will be interesting to explore the extent to which our results generalize beyond ImageNet, but we feel the ImageNet results alone are a distinct, clear and important milestone, and the extension to further datasets is beyond the scope and focus of this paper.
>
> 3- **Other approaches to fine-tuning:** Now that we know one can fine-tune a foundation text to image model to obtain powerful generative models for generative data augmentation, it is natural to ask whether different approaches to fine-tuning would be similarly effective or even better. We view this as a natural direction for further research, but the particular form of fine-tuning is not the primary issue in this paper, and hence an evaluation of myriad other fine-tuning methods is somewhat tangential to the main questions we address.
>
> 4- **Negative impact of large amounts of synthetic data:** The observation that synthetic data, in sufficient quantities, has a negative impact on classification performance is not particularly surprising.  This result was shown in Ravuri and Vinyals’ seminal 2019 Classification Accuracy Score paper.  What is surprising, to us, is that one can significantly improve the classification performance of SOTA models and training procedures with as much as 4M synthetic data points.  When mixing sufficiently large amounts of synthetic data with real data, with no other changes to architecture or training procedure, one would expect performance to approach that of models trained solely on synthetic data.  Unless the divergence between the generative model and the empirical data distribution is very close to zero, models trained on synthetic data alone are not expected to outperform. As shown with CAS experiments, current generative models do not meet that high level of performance. We do have ideas on how one might overcome this bottleneck, but this is the subject of on-going research.
>
> 5- **Other types of generative models:**  We do focus on diffusion models.  We do not explore other generative models in any depth, but we do report baselines with other families of generative models. As discussed in Section 2, this includes  Big-GAN-deep, VQ-VAE-2, and latent diffusion models that use prompt augmentations for generating synthetic data for Imagenet classifier training (Sariyildiz et al, 2022)

---

### Decision · Action_Editors · 2023-10-06

**Recommendation:** Accept as is

**Comment:**

This paper proposes to leverage large generative models with additional fine-tuning (here on ImageNet) on two dowstreams tasks.  First, they evaluate the quality of the samples of the fine-tuned model. Second, they perform downstream classification tasks on ImageNet using the generated samples / real samples and a mix of generated-real samples. They report degraded performance when the classification is performed only on generated data. However, when adding generated data to real data the classification results improve (up to a point). I found some of the findings quite interesting. Figure 6, showing the Top-1 and Top-5 classification accuracy of a ResNet-50 architecture trained on an union of synthetic/generated dataset shows evidence that too many synthetic examples might hurt the model. There is a consensus among the reviewers that this paper provides valuable insights to the future works.  While I share some of the concerns of Reviewer XfEJ regarding the hyperparameter selection, I think the authors successfully clarify these points during the discussion and in the revised manuscript.  For these reasons, I recommend the acceptance of the paper as is.

**Audience:**

This paper studies the capabilities of diffusion models. Answering such questions is key to broaden the applications of such models and is of great interest for the TMLR audience.

**Claims And Evidence:**

The claims of the paper are experimentally supported. The experiments are thorough and extensive.